# Entinostat in combination with nivolumab in metastatic pancreatic ductal adenocarcinoma: a phase 2 clinical trial

Pancreatic ductal adenocarcinoma (PDA) is characterized by low cytotoxic lymphocytes, abundant immune-suppressive cells, and resistance to immune checkpoint inhibitors (ICI). Preclinical PDA models showed the HDAC inhibitor entinostat reduced myeloid cell immunosuppression, sensitizing tumors to ICI therapy. This phase II study combined entinostat with nivolumab (PD1 inhibitor) in patients with advanced PDA (NCT03250273). Patients received entinostat 5 mg orally once weekly for 14-day lead-in, followed by entinostat and nivolumab. The primary endpoint was the objective response rate (ORR) by RECIST v1.1. Secondary endpoints included safety, duration of response, progression free-survival and overall survival. Between November 2017 and November 2020, 27 evaluable patients were enrolled. Three showed partial responses (11% ORR, 95% CI, 2.4%-29.2%) with a median response duration of 10.2 months. Median progression-free survival (PFS) and overall survival (OS) were, respectively, 1.89 (95% CI, 1.381-2.301) and 2.729 (95% CI, 1.841-5.622) months. Grade ≥3 treatment-related adverse events occurred in 19 patients (63%), including decreased lymphocyte count, anemia, hypoalbuminemia, and hyponatremia. As exploratory analysis, peripheral and tumor immune profiles changes were assessed using CyTOF, mIHC, and RNA-seq. Entinostat increased dendritic cell activation and maturation. Gene expression analysis revealed an enrichment in inflammatory response pathways with combination treatment. Although the primary endpoint was not met, entinostat and nivolumab showed durable responses in a small subset of PDA patients. Myeloid cell immunomodulation supported the preclinical hypothesis, providing a basis for future combinatorial therapies to enhance clinical benefits in PDA.

Pancreatic ductal adenocarcinoma (PDA) is the third leading cancer-related cause of death in adults, with 50,550 estimated deaths in 2023 in the United States[1]. Most patients with PDA have advanced disease at diagnosis, with a dismal overall prognosis that has remained virtually unchanged for many decades. The mainstay of treatment of advanced PDA is cytotoxic chemotherapy, and patients who are refractory to first-line chemotherapy have limited therapeutic options[2]. PDA is characterized by a T cell suppressive immune microenvironment and

is not responsive to current immune-based therapies targeting T cell functionality. In the largest clinical trial to date of immune checkpoint inhibitor (ICI) therapy in PDA, the response rate to ICI monotherapy was 0%[3]. Novel combination approaches are needed to transform PDA into an immune-responsive tumor type.

Multiple preclinical and clinical studies have demonstrated that histone deacetylase inhibitors can alter the tumor microenvironment (TME) immunogenicity[4–6]. In preclinical models of renal and

✉e-mail: nazad2@jhu.edu

castration-resistant prostate cancer, a low dose of histone deacetylases inhibitor (HDACi), entinostat, a class I/III HDAC inhibitor, in combination with IL-2 therapy or a survivin-based vaccine, inhibited tumor growth, reduced infiltrating regulatory T cells (Tregs) and increased the T effector (Teff) response[7]. Combination therapy of entinostat and anti-PD1 therapy in renal cell and lung cancer animal models also revealed the anti-tumor synergy of the combination[8]. Preclinical and clinical studies have suggested that the mechanism of immunomodulation for HDAC inhibitors is transcriptional reprogramming of some myeloid subsets to less T cell suppressive states. Orillion et al. demonstrated decreased myeloid-derived suppressor cells (MDSC) immunosuppressive function on T cells in co-culture assays in Lewis Lung Cancer models, which translated to improved survival of animals treated with HDAC inhibition as well as ICI therapy[9]. In the lung, breast, and colon cancer in vivo models, multiple groups have reported that combination therapy of ICI and HDAC inhibition decreases tumor growth and increases survival, with an impact on both numbers of granulocytic myeloid-derived suppressor cells (G-MDSCs) in the TME[9] and conversion on TAMs to favor a Th1 transcriptional state[10–12]. Translation of these preclinical data into patients has been compelling, with clinical trials of entinostat and anti-PD1 therapy reporting durable clinical responses in non-small cell lung cancer and melanoma patients resistant to anti-PD1 therapy. These studies further support the notion that HDACi may convert immunotherapy-resistant cancers into responders[13,14]. Correlative analyzes of these previous trials showed decreased monocytic MDSCs (M-MDSCs) in the periphery of responders.

Prior work from our group investigated the ability of HDACi to alter the suppressive function of suppressive immune cells in favor of recruiting T cells into PDA[15,16]. Using immune-competent syngeneic murine models, our previous studies demonstrated that i) the HDACi entinostat shifted the predominant M-MDSC population to a less immunosuppressive G-MDSC population, ii) entinostat exposure altered G-MDSC function converting them to a less immunosuppressive phenotype; iii) combination therapy of entinostat plus anti-PD-1 significantly improved survival as compared with mice treated with either agent alone[14,15]. Based on these data, we hypothesized that HDACi therapy could modulate the PDA TME, converting PDAs from a T cell-excluding cancer type into a T cell-permissive one.

Here, we report here a phase 2 clinical trial of entinostat and the PD-1 inhibitor, nivolumab, in previously treated metastatic PDA patients. This study demonstrates the safety and tolerability of the combination. In addition, although the primary endpoint was not met, three patients had a partial response by RECIST v1.1 criteria; the median duration of responses was 10.2 months, supporting further development of this approach for PDA treatment.

## Results

### A small subset of advanced PDAC patients responded to entinostat and nivolumab

**Patient characteristics.** From November 2017 to November 2020, a total of 30 patients with unresectable or metastatic, previously treated PDA were enrolled at the Johns Hopkins Sidney Kimmel Comprehensive Cancer Center in Baltimore, MD (Fig. 1). The baseline clinical characteristics and disease status of the patients are summarized in Table 1. It was a heavily pre-treated patient population, with 63% of patients (n = 19) having received ≥2 prior lines of systemic therapy (range 1–3).

**Clinical responses to the combined checkpoint and HDAC inhibition.** Of the 30 enrolled and treated patients, three discontinued study therapy due to toxicity (fatigue, anorexia and biliary sepsis, respectively) without evidence of disease progression before completing the first restaging scan (Fig. 2a,). Per protocol, after the first 13 patients were evaluable, an interim efficacy analysis showed that the study met

the criteria to move to the second stage of enrollment (one responder among the first 13 patients enrolled), and the trial was fully enrolled. In the evaluable patient population (n = 27, three patients were not evaluable as patients came off the study due to toxicity before completing the first cycle of treatment without evidence of disease progression.), the objective response rate (ORR) was 11.1% (3 of 27; 95% CI, 2.4–29.2%, p = 0.15). For the three patients with an objective response, the median duration of response was 10.2 months. Median progression-free survival (PFS) and overall survival (OS) were short, respectively 1.89 (95% CI, 1.381–2.301) and 2.729 (95% CI, 1.841–5.622) months and consistent with historical controls (Fig. 2e, f). PFS rates at 6 months and 12 months were 0.067 (95% CI, 0.017- 0.254), and 0.067 (95% CI, 0.017–0.254). All patients progressed by 24 months. Twenty-four of the 27 patients (89%) had an elevated pancreatic cancer marker, CA19.9, at the study baseline, and CA19.9 levels were followed routinely for 19 of these patients. A waterfall plot of maximum CA19.9 changes on study therapy for all subjects with an elevated CA19.9 at baseline and correlation with the radiological response is shown in Supplementary Fig. 1.

A summary of the observed clinical responses for the three patients that achieved partial response is shown in Fig. 3. Of note, all 3 of these responders also experienced immune-related adverse events (irAE). Responder 1 (patient P28) presented with metastatic disease to the lungs and lymph nodes after neoadjuvant and adjuvant therapy for stage III disease. The tumor was microsatellite stable (MSS) and showed a low tumor mutation burden (TMB; 4.41 mutations/Mb). The patient initiated treatment with entinostat and nivolumab with a partial response (PR) and normalization of CA19.9 at two months and a complete response (CR) of the lung metastasis at seven months. The patient's treatment was complicated by immune-related renal tubular acidosis requiring electrolyte replacement, and the patient elected to stop treatment after 9 cycles. The cancer continued to respond, and ultimately, the patient passed away due to elective surgery complications (unrelated to cancer) without evidence of disease progression 10.2 months after initiating therapy.

Responder 2 (patient P18) presented with MSS, low TMB (3.53 mut/Mb) PDA metastatic to liver and peritoneum, progressing despite two lines of standard chemotherapy. The patient subsequently

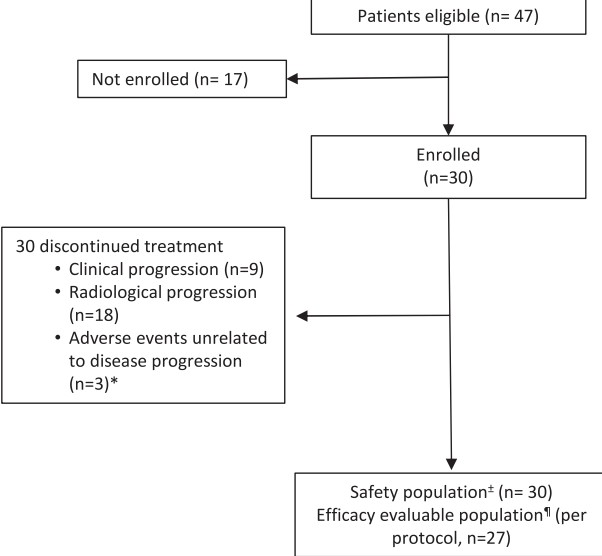

**Fig. 1 | Consort diagram of enrolled patients. n values represent the number of patients at each stage of the protocol.** ±The safety population included all patients who received one or more doses of entinostat. The efficacy evaluable population included all patients in the safety population with measurable disease at baseline per RECIST v1.1. Three patients were not evaluable as patients came off the study due to toxicity before completing the first cycle of treatment without evidence of disease progression.

## Table 1 | Patient characteristics

| Patient Characteristic, N = 30 | |
|---|---|
| On Study Age (years), median | 63.5 (32–79) |
| **Gender, N (%)** | |
| Female | 13 (43) |
| Male | 17 (57) |
| **Race, N (%)** | |
| Black, | 2 (7) |
| White | 28 (93) |
| **Ethnicity, N (%)** | |
| Hispanic or Latino | 1 (3) |
| Not Hispanic or Latino | 27 (90) |
| Unknown | 2 (7) |
| **Location of Primary Tumor, N (%)** | |
| Head | 15 (50) |
| Body/tail | 1 (3) |
| Body | 5 (16) |
| Tail | 8 (27) |
| Uncinated | 1 (3) |
| **Differentiation, N (%)** | |
| Poorly differentiated | 14 (47) |
| Moderate differentiated | 11 (37) |
| Well-differentiated | 1 (3) |
| Unknown | 4 (13) |
| **Stage at Diagnosis, N (%)** | |
| IB-IIB | 12 (40) |
| III | 3 (10) |
| IV | 15 (50) |
| **Surgical Resection, N (%)** | |
| N | 19 (63) |
| Y | 11 (37) |
| ECOG 0, N (%) | 8 (27) |
| **Lines of Chemo, N (%)** | |
| 1 | 11 (37) |
| 2 | 17 (57) |
| 3 | 2 (7) |
| **Metastatic site at enrollment, N (%)** | |
| Liver | 26 (87) |
| Lung | 15 (50) |
| Peritoneum | 13 (40) |
| Lymph nodes | 11 (37) |
| Other (adrenal gland, bone) | 3 (10) |
| CA 19-9 secretors, N (%) | 27 (90) |

enrolled in the trial and received entinostat plus nivolumab, resulting in a maximum response of −64% and normalizing his CA19.9. Despite stopping treatment after approximately 1 year due to grade 3 itching, the patient maintained an objective response for nearly two years before oligo-progression in the adrenal gland. The patient then underwent local radiation therapy and showed continued response in other disease sites for 7 additional months. This patient remains alive at the time of this report, 5 years after initiating therapy.

Responder 3 (patient P38) was a patient with stage IV PDA (MSS, TMB 13.23 mut/Mb) with metastasis to the liver and lymph nodes. The patient achieved a maximum response of −43% on third-line treatment within the clinical trial. Of note, after three cycles, the patient developed grade 3 colitis, for which treatment with steroids, tapered over 6 weeks, was started; the patient then continued on trial, and scans at

6 months from starting the clinical trial showed a mixed response with regression of the target lesions and progression of one left supraclavicular lymph node. This patient came off study and achieved a partial response upon the next-line standard of care treatment.

### Safety of combination therapy

Toxicity was evaluated in all the patients who received at least one dose of entinostat ($n = 30$). The safety and toxicity of entinostat and nivolumab in combination were consistent with the toxicity profile of the two individual drugs and with the prior experience of this combination in other tumor types[17].Twenty-seven (90%) patients had AEs related to one of the study drugs; 27 (90%) patients experienced entinostat-related AEs, and 14 (47%) patients experienced nivolumab-related AEs (Table 2). Six patients (20%) required entinostat dose reduction (fatigue $n = 3$, and anorexia, neutropenia and thrombocytopenia one each respectively). Grade ≥3 treatment-related AE (TRAEs) were encountered in 19 (63%) patients and were more commonly attributed to entinostat ($n = 18$, 95%) than to nivolumab ($n = 4$, 21%). The most common grade ≥3 AEs were decreased lymphocyte count and anemia. No grade 5 adverse events were observed.

### Entinostat reprograms tumor-associated and peripheral myeloid populations in PDA, promoting a T cell-permissive TME

**Explorative immune TME analysis by mIHC.** Embedded correlative biospecimen analysis of serial tissue and blood was a key component of the study to gain a deeper understanding of the effect of entinostat on the PDA TME. Multiplexed immune histochemistry (mIHC) staining for 23 antibodies focused on characterizing the TME was performed on serial tumor biopsy specimens obtained at baseline ($n = 26$), after two weeks of entinostat lead-in therapy at C1D1 ($n = 21$), and at week 6 following entinostat given with nivolumab on combination therapy (C2D1, $n = 4$) to comprehensively assess the effects of therapy on the TME (Supplementary Fig. 2).

We calculated the median population of granulocyte, macrophage, and monocyte abundance (cells/mm$^2$) and found the baseline samples of responders trended toward a greater myeloid cell density at baseline in the total analyzed areas, although statistical significance was not reached (Supplementary Data 1 and 2). Furthermore, to measure monocyte/macrophage phenotype, we examined a ratio of CD163 and CD68. We found decreased percentage of CD163$^+$/CD68$^+$ macrophages after two weeks of entinostat lead-in therapy (C1D1) (Supplementary Fig. 3a,b and Supplementary Data 3 and 4). This trend suggested a "skewing" toward a more favorable Th1-like phenotype.

Dendritic cells (DCs) and macrophages contribute to adaptive immunity through antigen presentation and priming and conditioning effector cells. Notably, upon treatment with entinostat, we observed a trend of increased ratio of DC-LAMP$^+$ mature DCs (with the expression of lysosome-associated membrane protein 3, LAMP-3) to DC-LAMP$^-$ immature DCs (Supplementary Fig. 3c and Supplementary Data 5).

**CyTOF profiling of the PBMCs.** Intratumoral myeloid cells are often recruited from the periphery, indicating crosstalk between the innate and adaptive immune response. To evaluate the systemic immunologic effects related to entinostat alone or in combination with nivolumab, we conducted an exploratory analysis to assess the peripheral immune profiles at baseline and post-treatment using high-dimensional mass cytometry by time of flight (CyTOF) with a myeloid cell-oriented antibody panel (Supplementary Data 6) on cryopreserved PBMC samples isolated from the blood of a total of 28 patients for which baseline and post-entinostat treatment samples were available for analysis. We examined the myeloid cellular components, including pre-Dendritic Cell (preDC)/classic monocytes (Lin$^-$CD14$^+$CD16$^-$), which are early progenitors to mature dendritic cells, non-classical monocytes (Lin$^-$CD14$^{low}$CD16$^+$), early myelocytes, plasmacytoid dendritic cells (pDC) and classical dendritic cells (cDCs)[18,19].

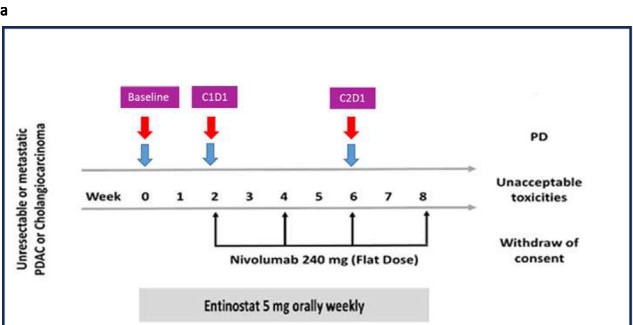

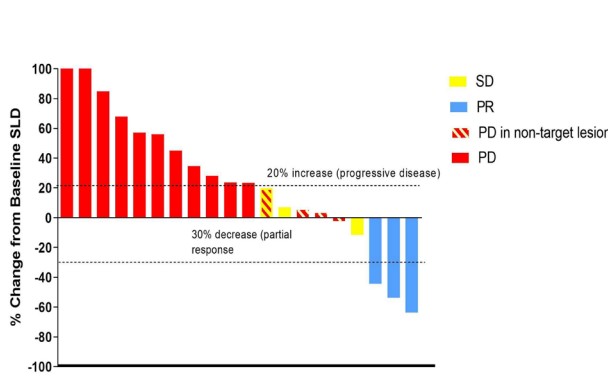

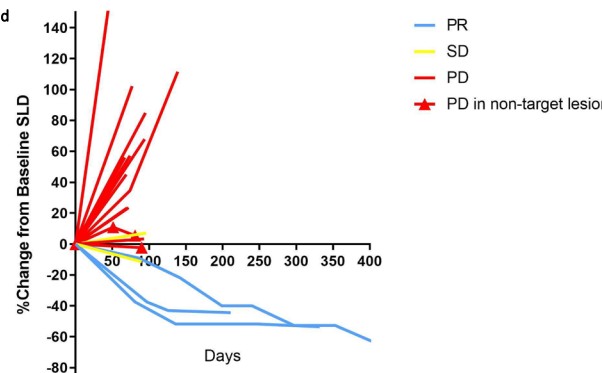

**b**

| Efficacy | | |
|---|---|---|
| | ITT (n= 30) | Per Protocol (n= 27) |
| **ORR** | 3 (10%) | 3 (11%) |
| SD | 2 (7%) | 2 (7%) |
| PR | 3 (10%) | 3 (11%) |
| **DCR** | 5 (17%) | 5 (18.5%) |
| **Discontinued[a]** | 10 (33%) | 7 (26%) |

[a]Discontinuation before the first restaging scan was due to toxicity (3), or clinical progression (4).

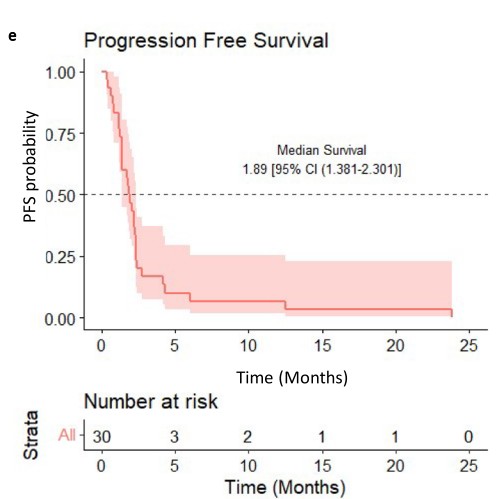

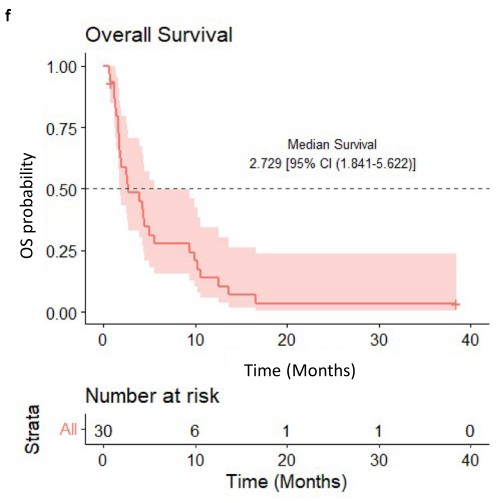

**Fig. 2 | Clinical responses to entinostat and nivolumab.** a Study schema. Tumor biopsy (blue arrow) and blood (red arrow) collection timepoints; Baseline, C1D1 (after 2-week entinostat Run-in), C2D1 (after 6 weeks of combination therapy with entinostat + nivolumab). **b** Efficacy by best overall response by RECIST 1.1, shown are the patients that were enrolled for intention to treat (ITT) and those who received at least one CT scan RECIST reading to assess primary endpoint (per protocol). **c** The change from baseline in the target lesion diameter according to Response Evaluation Criteria in Solid Tumors (RECIST), version 1.1, for all evaluable patients (n = 20 patients). **d** Spider plot of radiographic responses to treatment with entinostat plus nivolumab, for all evaluable patients (n = 20 patients). Tumor responses were measured at regular intervals, and the values shown are the largest percentage change in the sum of the longest diameters from the baseline measurements of each measurable tumor. Each line represents one patient. **e**, **f** Kaplan–Meier curves of PFS (**e**) and OS (**f**). The 95% CIs for point estimates are shown in red shading. Source data are provided as a Source Data file. *DCR, disease control rate; ITT, intention to treat; ORR, overall response rate; OS, overall survival; PD, progressive disease; PFS, progression free survival; PR, partial response; SD, stable disease; SLD, sum of longest diameters of the target lesions.*

The latter were further classified into two functionally distinct lineages: CD103[+] migratory cDC1 and CD11b[+] cDC2 (Figs. 4a, b). We found a statistically higher abundance of circulating preDC/classical monocytes after entinostat treatment compared to baseline, while pDCs, cDC2, and early myelocytes were decreased (Fig. 4c and Supplementary Data 7 and 8). When we examined in detail the functional phenotypes of these populations, we found greater expression levels of

CLEC9A in cDC1 and CD103 in cDC2 (Fig. 4d), respectively, while Arginase-1 expression was significantly decreased in non-classical monocytes (Fig. 4d). Significantly increased expression levels of CD40, CCR5, and CCR2 (Fig. 4e–g and Supplementary Data 9 and 10) in several subtypes of monocytes, myelocytes, and DCs were also observed after entinostat monotherapy. Collectively, the results indicate increased DC maturation and antigen presentation/processing as

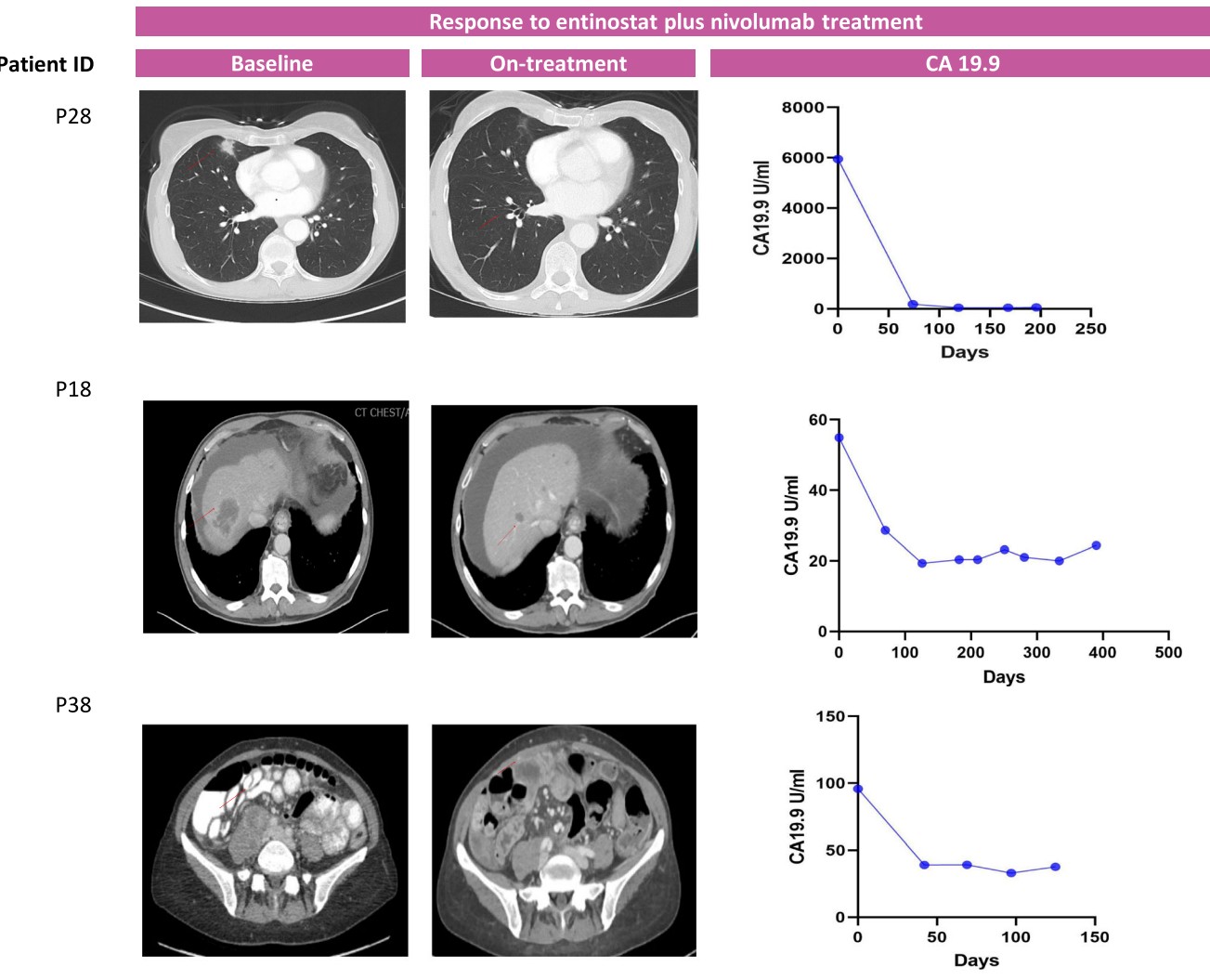

**Fig. 3 | Radiologic scans and change in CA19.9 of the three responders during the course of entinostat plus nivolumab therapy.** CA 19.9 (Reference range: 0.0–36.0 U/mL).

a consequence of HDAC inhibition. In addition, a significant increase in PD-L1 and PD-L2 expression in several subtypes of monocytes, myelocytes, and DCs was observed after entinostat exposure in the periphery (Supplementary Fig. 4a,b and Supplementary Data 9) suggesting a potential IFNγ-related T cell response associated with entinostat therapy.

**Explorative analysis of plasma cytokine and chemokine profiles.** Our hypothesis that entinostat modulates the myeloid compartment was also supported by measurements by Luminex of chemokines in the plasma (Supplementary Data 11 and 12), which revealed downregulation of MCP-1 (CCL2, $p = 0.007$), soluble sCD40L ($p = 0.024$), Eotaxin (CCL11, $p = 0.033$), MDC ($p = 0.033$) and MIP-1B (CCL4, $p = 0.046$) in post-entinostat samples (Fig. 5a and Supplementary Data 13). Mounting evidence indicates that numerous effects of these chemokines are pro-tumorigenic, favoring recruitment of Th2-like monocytes into tumors, serving as a chemoattractant for Tregs that inhibit T cell effector functions[20,21] and switch T cell differentiation towards Th2-like cells[22–25]. Consistently, we observed upregulation of several proinflammatory cytokines, e.g., IL-15 ($p = 0.016$)[23–25], which have been associated with eosinophils expansion, activated myeloid and NK cells, and T proliferation and cytotoxic actions[26–28]. However, we observed some changes that did not associate with improved immune responses[29,30], e.g., lower IP-10 (also known as CXCL10,

$p = 0.002$) and higher IL-6 ($p < 0.001$)[26,27] (Fig. 5b and Supplementary Data 12 and 13). These changes were reverted at week 6 following entinostat given with nivolumab (C2D1), suggesting the importance of combining entinostat therapy with anti-PD1 to maximize the anti-tumor immune response.

**Phenotypic and functional changes within the circulating T cell pool following entinostat plus nivolumab**
**Explorative immune TME analysis by mIHC.** Substantial evidence supports the notion that TAMs subvert tumor-infiltrating T lymphocyte function, thus restraining the efficacy of ICIs[31]. We hypothesized that reprogramming the tumor myeloid compartment through HDAC inhibition prior to giving anti-PD-1 immunotherapy might facilitate CD8$^+$ T cell tumor infiltration and anti-tumor activity. Our data revealed that the relative frequencies of major T cell subsets were generally similar at baseline compared with after entinostat treatment (Supplementary Fig. 5a). However, treatment with entinostat monotherapy was associated with a decreased proliferation (Ki67$^+$) in CD4$^+$ T cells and reduced proliferation in other T cell subsets from baseline in the overall study population including a decrease in Th2, Th0, Th9, and Th17 CD4$^+$T cells (Supplementary Fig. 5b and Supplementary Data 3 and 4), which are known to negatively affect the anti-tumor response through cytokine and chemokine expression leading to multiple mechanisms of effector T cell inhibition.

**Table 2 | Treatment-related adverse events in patients**

| | Grade 1–2 N | Grade 3–4 N | Related to Entinostat | Related to Nivolumab | Total N = 30 N (%) |
|---|---|---|---|---|---|
| **Blood and lymphatic system disorders** | | | | | |
| Anemia | 1 | 8 | Yes | No | 9 (30%) |
| Febrile neutropenia | 0 | 1 | Yes | No | 1 (3%) |
| Lymphocyte count decreased | 0 | 8 | Yes | No | 8 (27%) |
| Neutrophil count decreased | 1 | 3 | Yes | No | 4 (13%) |
| Platelet count decreased | 2 | 1 | Yes | No | 3 (10%) |
| **Gastrointestinal disorders** | | | | | |
| Colitis | 0 | 1 | No | Yes | 1 (3%) |
| Constipation | 1 | 0 | Yes | No | 1 (3%) |
| Diarrhea | 3 | 0 | Yes | Yes | 3 (10%) |
| Dry mouth | 2 | 0 | Yes | No | 2 (7%) |
| Dysgeusia | 2 | 0 | Yes | Yes | 2 (7%) |
| Early satiety | 1 | 0 | Yes | No | 1 (3%) |
| Flatulence | 1 | 0 | No | Yes | 1 (3%) |
| Gastro-esophageal reflux | 1 | 0 | Yes | No | 1 (3%) |
| Mucositis, oral | 4 | 0 | Yes | Yes | 4 (13%) |
| Nausea | 10 | 0 | Yes | No | 10 (33%) |
| Vomiting | 6 | 0 | Yes | No | 6 (20%) |
| **General disorders** | | | | | |
| Chills | 2 | 0 | Yes | Yes | 2 (7%) |
| Edema lower extremities | 4 | 0 | Yes | No | 4 (13%) |
| Fatigue | 20 | 0 | Yes | Yes | 20 (67%) |
| Fever | 3 | 0 | No | Yes | 3 (10%) |
| Gait disturbance | 1 | 0 | No | Yes | 1 (3%) |
| **Infections and infestations** | | | | | |
| Common cold | 1 | 0 | No | Yes | 1 (3%) |
| Lung infection, pneumonia | 0 | 1 | No | Yes | 1 (3%) |
| Thrush | 4 | 0 | Yes | No | 4 (13%) |
| **Liver disorders** | | | | | |
| ALT, elevated | 1 | 0 | Yes | Yes | 1 (3%) |
| AST, elevated | 1 | 0 | Yes | Yes | 1 (3%) |
| Bilirubin, increased | 1 | 0 | Yes | Yes | 1 (3%) |
| **Metabolism and nutrition disorders** | | | | | |
| Anorexia | 15 | 0 | Yes | No | 15 (50%) |
| Hypoalbuminemia | 0 | 3 | Yes | No | 3 (10%) |
| Hypokalemia | 0 | 1 | Yes | No | 1 (3%) |
| Hyponatremia | 0 | 5 | Yes | No | 5 (17%) |
| Weight loss | 2 | 0 | Yes | No | 2 (7%) |
| **Musculoskeletal and Connective Tissue Disorders** | | | | | |
| Arthritis | 1 | 1 | No | Yes | 2 (7%) |
| **Nervous System Disorders** | | | | | |
| Paresthesia | 1 | 0 | Yes | No | 1 (3%) |
| **Renal and urinary disorder** | | | | | |
| Renal tubular acidosis | 1 | 0 | Yes | Yes | 1 (3%) |
| Creatinine, elevated | 1 | 0 | No | Yes | 1 (3%) |

**Table 2 (continued) | Treatment-related adverse events in patients**

| | Grade 1–2 N | Grade 3–4 N | Related to Entinostat | Related to Nivolumab | Total N = 30 N (%) |
|---|---|---|---|---|---|
| **Skin and subcutaneous tissue disorders** | | | | | |
| Pruritis | 0 | 1 | No | Yes | 1 (3%) |
| Rash | 3 | 1 | No | Yes | 4 (13%) |

**CyTOF profiling of the PBMCs.** We quantified activation, proliferation, and other functional markers on CD4+ and CD8+ effector T cell populations in the periphery by CyTOF using a lymphoid cell-oriented panel (Fig. 6a, b). Entinostat led to a significant increase in the number of effector memory cells (ThEM; CD62L⁻CD44⁺) and central memory T cells (ThCM; CD62L + CD44 + ) (Figs.6c, d). Upon exposure to the entinostat plus nivolumab combination, decreased naïve T cell subtypes (ThN) were observed (Fig. 6d and Supplementary Data 14 and 15). These phenotypic and functional changes within the circulating T cell compartment suggest that entinostat may expand polyfunctional, antigen-experienced, and effector memory T cell responses, which may translate into durable responses for some patients. The population of circulating B cells was also decreased in post-entinostat samples (Supplementary Fig. 6), while no significant differences in intratumoral B cell infiltration were found (Supplementary Data 1–4). We also investigated if entinostat influences the phenotype of these B cells and observed that CD19, CCR2, CD40, as well as CD11c expression were increased in post-entinostat samples (Supplementary Fig. 6b and Supplementary Data 9 and 10).

## Gene expression alterations in the microenvironment

We also conducted an exploratory, post-hoc analysis to investigate whether HDAC inhibition, with or without nivolumab, was associated with distinct transcriptional signatures in the context of therapy. We performed whole transcriptome RNA sequencing (RNA-seq) on serial tumor samples at baseline ($n = 23$), after 2 weeks of entinostat monotherapy (C1D1, $n = 18$), and at week 6 following entinostat given with nivolumab (C2D1, $n = 4$). We did not find significant transcriptional changes at the 2-week entinostat monotherapy lead-in (Supplementary Fig. 7, Supplementary Data 16), likely due to the known delayed effects of these epigenetic inhibitors on gene expression and the early timing of these biopsies. However, 2 weeks was the longest time that the study team felt patients could safely receive a priming treatment as a lead-in that does not have anti-tumor activity as a single agent. Significantly more transcriptional differences were noted at the 6-week biopsy time-point with a combination of entinostat and nivolumab therapy (Fig. 7a, b and Supplementary Data 17 and 18).

We found 383 and 346 significantly differentially expressed genes in baseline vs C2D1 and C1D1 (after two weeks of entinostat lead-in therapy) vs C2D1 comparisons, respectively (Fig. 7a, b and Supplementary Data 17 and 18). Three hundred genes were upregulated when nivolumab was combined with entinostat at C2D1 comparing to entinostat alone (C1D1, Fig. 7b and Supplementary Data 19), many of which were immune checkpoint response as well as antigen-presenting cell (APC)-associated genes. These include many *HLA* genes, *CXCL9* and *CXCL10*, *CCR2*, *CCR3*, *CTLA4*, and *CD274* (PD-L1). Consistent with this observation, combined entinostat plus nivolumab significantly upregulated inflammatory response and IFN gamma and alfa inflammasome signaling pathways (Fig. 7c and Supplementary Data 19 and 20). Notably, entinostat treatment alone during the lead-in phase significantly upregulated the TGF beta pathway and downregulated the MYC oncogene pathway (Fig. 7c and Supplementary Data 21).

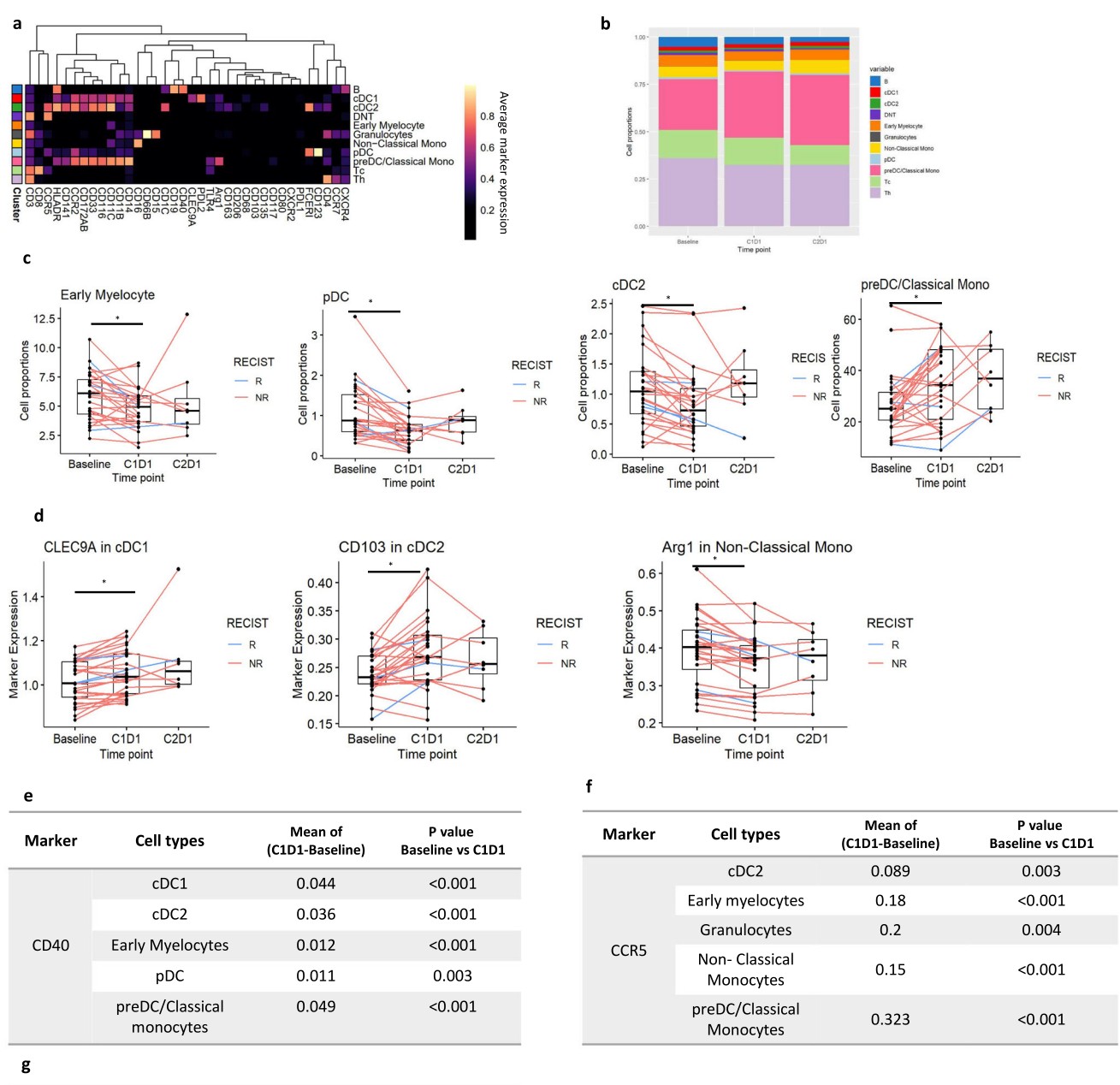

**Fig. 4 | Entinostat treatment altered abundance of myeloid populations in PBMC. a** Heatmap of the normalized expression for all markers used during FlowSOM from baseline and post-treatment samples using a Myeloid cell-oriented oriented CyTOF panel. *N* = 28 paired patient sample between baseline and C1D1, and *n* = 8 paired samples between C1D1 and C2D1. Cell subtypes listed on the right were assigned based on differential expression of the markers listed across the bottom. B cells, cDC1- classical dendritic cell type 1, cDC2-classic dendritic cell type 2, DNT- double negative T cells (neither CD4 nor CD8 expressing T cells), early myelocytes, Granulocytes, Non-classic monocytes, pDC-plasmacytoid dendritic cell, pre-DC - preDendritic cells (preDC)/Classic monocytes, Tc - Cytotoxic T cells, Th- Helper Tcells. **b** Bar plot of the proportion of myeloid cell populations at baseline, after Entinostat lead-in (C1D1), and post combination (C2D1). **c** Proportions of preDC/Classic monocytes (p = 0.007); early myelocytes (*p* = 0.001), pDC (*p* < 0.001), and cDC2 (*p* < 0.001) at different time points. Each line corresponds to one patient. Red line corresponds to a non responder; blue line

corresponds to a responder. **d** Evaluation of CLEC9A (*p* = 0.016), CD103 (*p* < 0.001), and Arginase (*p* < 0.001) expression across cellular groups in the myeloid partition at given time points. **e–g**, Evaluation of CD40, CCR5, and CCR2 expression across cellular groups in the myeloid partition at given time points *P*-values from paired two-sided Wilcoxon test between time points and the mean of the difference between time points are shown. Statistically significant *P* value is shown as follows: **P* < 0.05. For all panels: *N* = 28 paired patient sample between baseline and C1D1, and *n* = 8 paired samples between C1D1 and C2D1. Box plots show the median and upper and lower quartiles and whiskers extend to 1.5× the interquartile range. Box plots show the median and upper and lower quartiles and whiskers extend to 1.5× the interquartile range. Source data are provided as a Source Data file. *IL-6, interleukin 6; IP10, interferon γ-induced protein 10 kDa; MCP-1, Monocyte Chemoattractant Protein-1; MDC, Macrophage-derived chemokine; MIP-1b, Macrophage inflammatory protein-1 beta; sCD40L, soluble CD40 Ligand; NR, non responders; R, responders.*

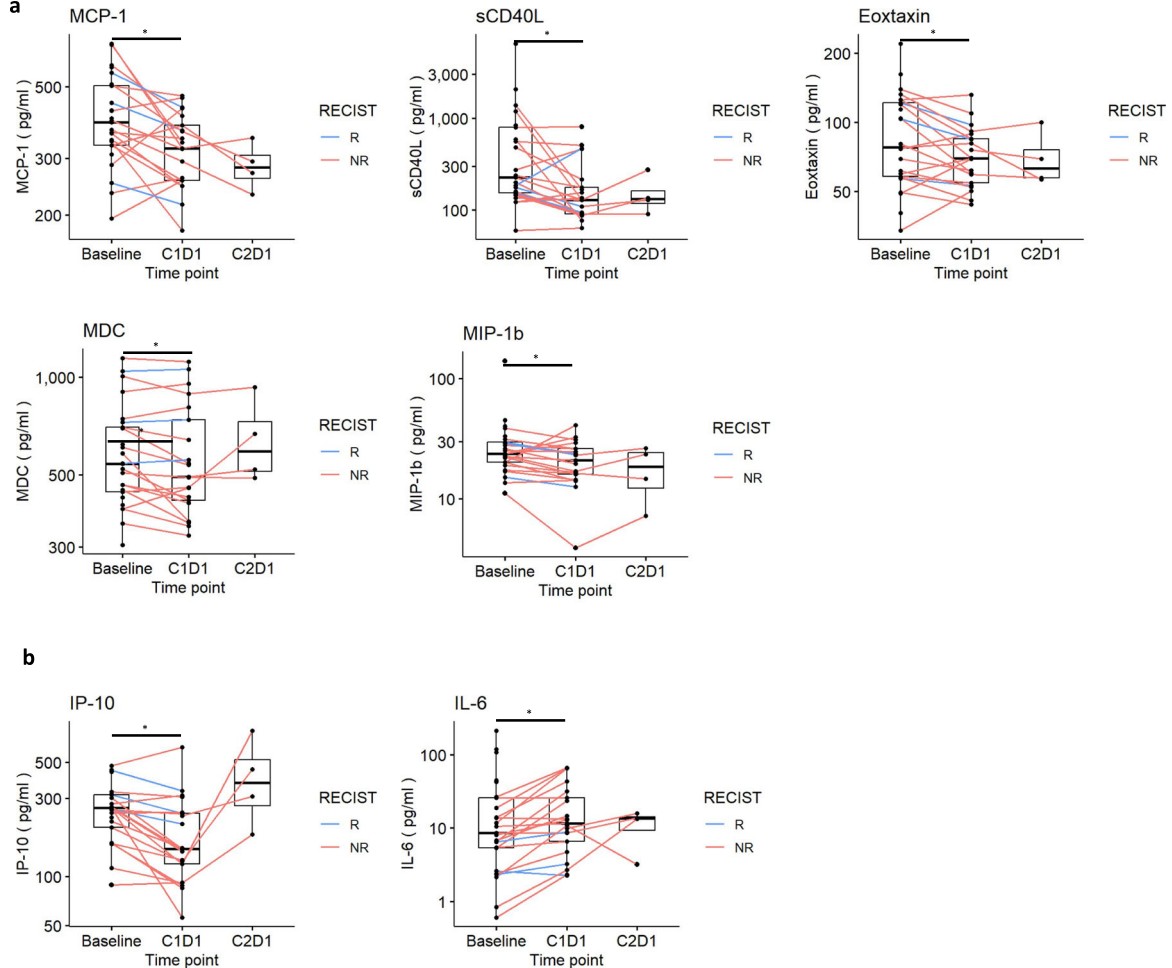

**Fig. 5 | Plasma concentration of circulating cytokines reveals downregulation of chemochines associated with TAM infiltration and angiogenesis upon entinostat exposure. a** change in plasma concentration of MCP-1 ($p = 0.007$), sCD40L ($p = 0.024$), Eoxtaxin (p = 0.033), MDC ($p = 0.033$), and MIP-1b (0.046); **b** change in plasma concentration of IP-10 ($p = 0.002$) and IL-6 ($p = <0.001$). *P*-values from paired two-sided Wilcoxon test between time points. Statistically significant *P* value is shown as follows: **P* < 0.05. *N* = 20 paired patient samples between baseline and C1D1 and and *n* = 4 paired samples between C1D1 and C2D1. Source data are provided as a Source Data file. *IL-6, interleukin 6; IP10, interferon γ-induced protein 10 kDa; MCP-1, Monocyte Chemoattractant Protein-1; MDC, Macrophage-derived chemokine; MIP-1b, Macrophage inflammatory protein-1 beta; sCD40L, soluble CD40 Ligand; NR, non responders; R, responders.*

## Discussion

Extensive prior research has shown that HDAC inhibition modulates the tumor immune microenvironment by decreasing the number of suppressive innate immune cells and altering their T cell suppressive function[7,32–34], resulting in complementary clinical activity with anti-PD1 immunotherapy. This clinical trial test HDAC inhibition and immune checkpoint inhibition in PDA patients. It provides initial evidence of safety, tolerability, and clinical activity in a subset of patients, with the combination in a subset of patients with an otherwise completely immune checkpoint-resistant cancer. Leveraging our preclinical models of PDA and clinical specimens from the trial, we addressed unanswered mechanistic questions about the effects of HDACi on the TME.

In this study, we performed explorative analysis to compare pre- and post-entinostat-conditioned tumor samples prior to immunotherapy in PDA patients. As predicted by our preclinical models, we observed a trend towards decreased CD163$^+$/CD68$^+$ macrophages in the post-entinostat biopsies, revealing a favorable reprogramming effect on the myeloid compartment by entinostat towards a less "immunosuppressive" phenotype of TAMs. TAMs are one of the most abundant immune cell populations in the PDA stroma and are often transcriptionally skewed toward Th2 deviation in the TME, which

supports several of their pro-tumoral activities, including T cell resistance, metastasis, as well as chemotherapeutic and immunotherapy resistance[35,36]. TAM density has also been reported as an independent prognostic factor in PDA patients and is associated with a higher risk of disease progression and shorter overall survival[37].

Second, we demonstrate that entinostat treatment promotes DC maturation within the tumor and induces CLEC9A, CD103, and CD40L expression in peripheral myeloid cells, indicating increased DC maturation and antigen processing, migration, and cross-presentation. PDA is characterized by the reduced number and function of DCs, which negatively impacts antigen presentation and contributes to immune tolerance[38]. Prior preclinical work has found that increasing the number and maturation of cDCs restores anti-tumor T cell immunity[39]. Levels of CLEC9A, a highly specific marker for mature cDC1, have correlated with improved survival in different cancer types[40], and previous reports have shown that high densities of intratumoral LAMP$^+$ DCs are associated with tertiary lymphoid structures, Th1 polarization, and cytotoxic activity[41,42]. Our work uncovers an effect of entinostat on DC differentiation and maturation that could overcome the 'semi-maturation' phenotype of DC in PDA, potentiating their role in orchestrating adaptive immune responses at the tumor site. These findings also indicate that combining HDAC inhibition with other immunomodulatory

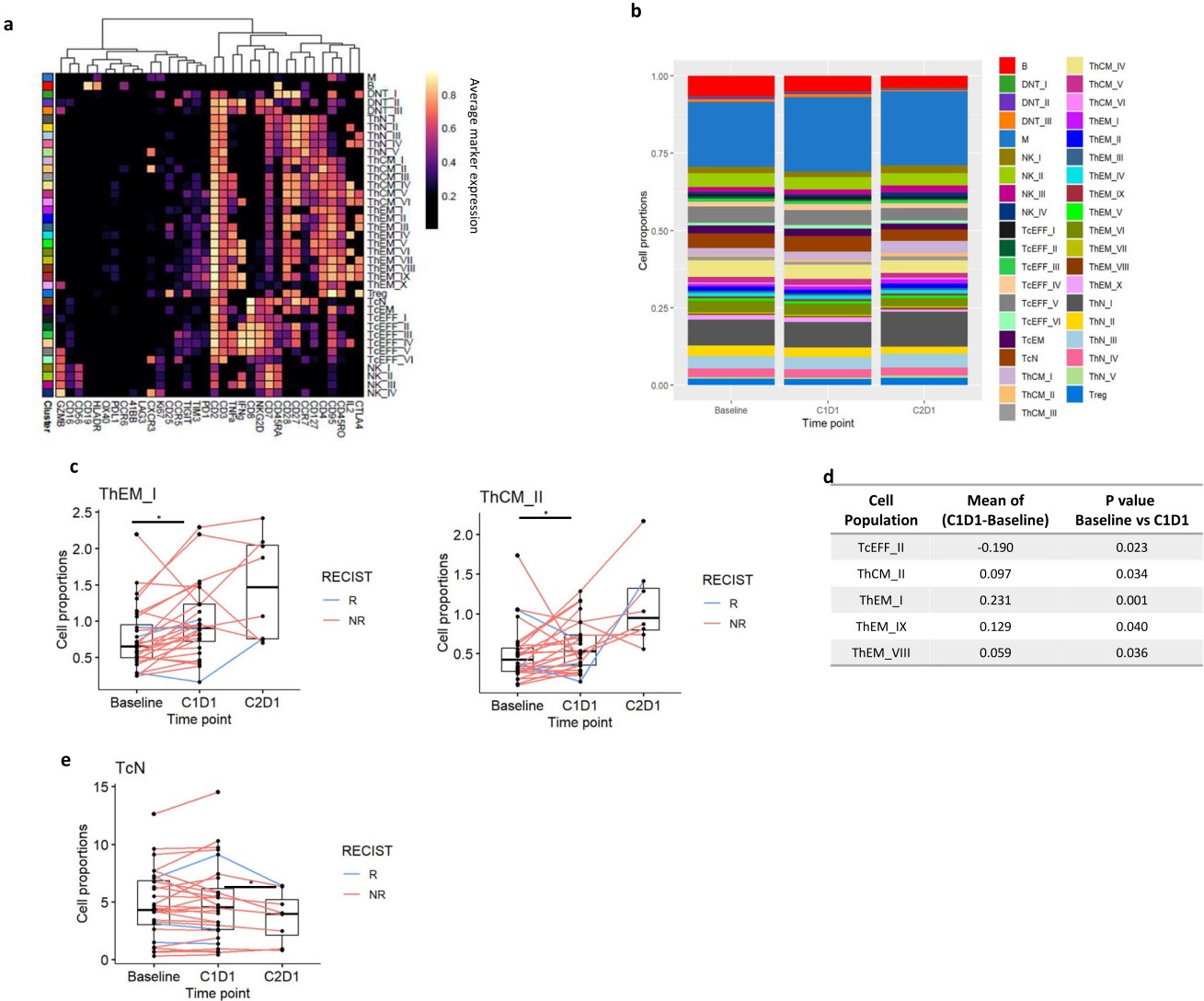

**Fig. 6 | Entinostat promotes differentiation toward memory away from naive in peripheral T cells and induces changes in functional states of immune cell subsets. a** Heatmap of the normalized expression for all markers used during FlowSOM from baseline and post-treatment samples using a lymphoid cell-oriented CyTOF panel. Cell sub types listed on the right were assigned based on differential expression of the markers listed across the bottom. **b** Bar plot of the proportion of T cell populations at different timepoint. **c,d** Graphs representing proportion of effector memory T (ThEM) cells (*p* = 0.001) and central memory (ThCM) T cells ($p = 0.03$) comparing baseline vs. post entinostat; **e** Graphs representing proportion of naïve T cell subtypes (TcN) cells comparing baseline vs. post entinostat ($p = 0.02$). *P*-values from paired two-sided Wilcoxon test between time points. Statistically significant *P* value is shown as follows: \**P* < 0.05. For all panels: *N* = 28 paired patient samples between baseline and C1D1, and *n* = 8 paired samples between C1D1 and C2D1. Box plots show the median and upper and lower quartiles and whiskers extend to 1.5× the interquartile range. See abbreviations of clusters are in the Source data table 7. Source data are provided as a Source Data file.

strategies that rely on DC function, such as cancer vaccines, could be even more effective and is ongoing research in our group.

Peripheral markers of response are sometimes informative of treatment-induced immune activity at the tumor site. After two single doses of entinostat, we observed increased polyfunctionality and central and effector memory circulating CD8⁺ T cells. These data suggest that entinostat may have an indirect role in activating naïve CD8 T cells to recognize their cognate antigen and proliferate, giving rise to a central memory CD8 T cell population, which has been postulated to be more potent on a per-cell basis in mediating antigen clearance compared to effector memory cells[43]. Therefore, the generation of central memory T cells should be an important immunologic endpoint to consider in future preventative and therapeutic vaccine trials. However, we did not observe a significant change in intratumoral T cell trafficking, perhaps due to the timing of sample collection or because these T cells proliferate in the peripheral blood or regional lymph nodes before they infiltrate into tumor sites.

Similar to other groups[44–46], we report that entinostat treatment led to significant upregulation of PD-L1/L2 expression. This observation further supports the need to add PD-1 inhibition to maximize the antitumor effects of entinostat, although more studies are needed to understand if early upregulation of PD1/L1 pathways could be used as a surrogate marker to predict response in this setting. We also showed that entinostat alone alters myeloid chemotaxis signaling with decreased expression of many immunosuppressive and pro-tumorigenic chemokines. Chemokines are involved in tumor-specific effector T cell trafficking, retention, and regulation of their in situ effector functions. We observed that chemokine changes that would be considered more pro-tumorigenic seemed reversed at week 6 upon nivolumab exposure, suggesting that HDACi both systemically and intratumorally, is well partnered with combination checkpoint inhibition.

We also observed a significant decrease in B cells in peripheral blood upon entinostat treatment. The role of B cells in anti-tumor

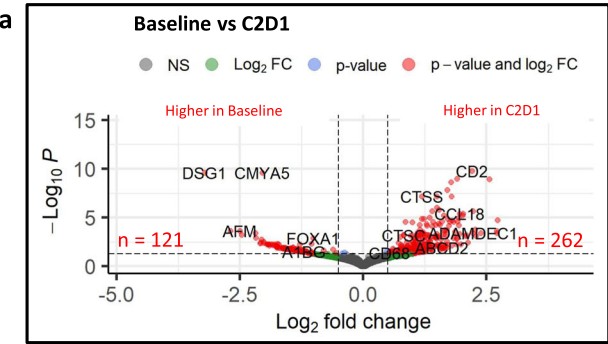

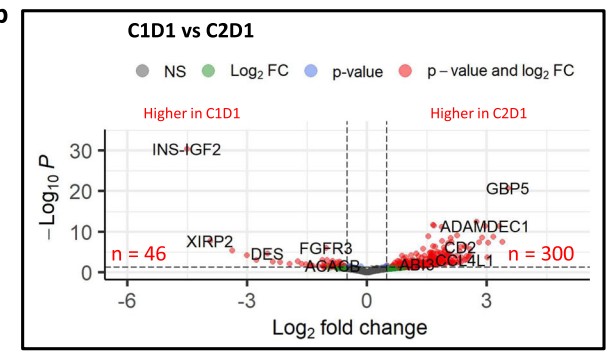

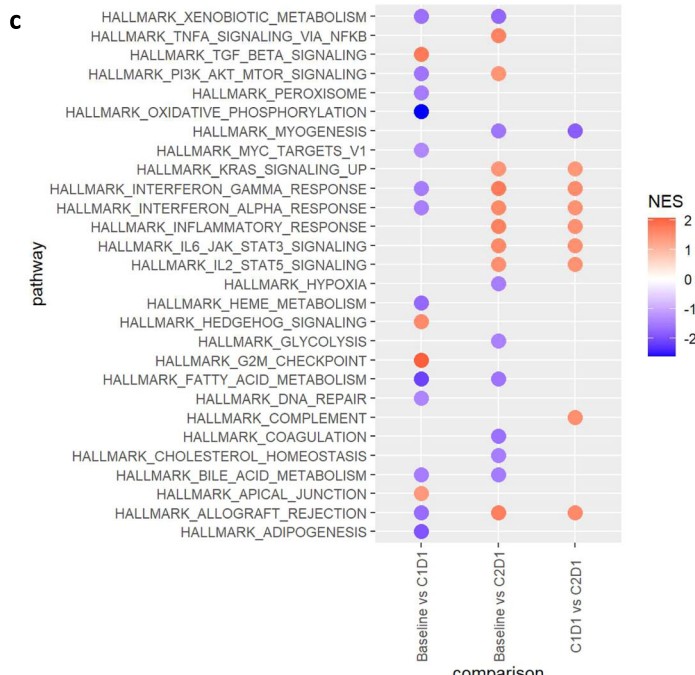

**Fig. 7 | RNA-seq analysis of differentially expressed genes between baseline and on treatment samples in this study cohort of PDA. a,b** Volcano plots indicating differentially expressed genes in the analysis of paired samples. The paired differential expression analysis was performed using the DESeq2 package (v1.32.0) comparing Baseline vs C2D1 samples (**a**) and C1D1 vs C2D1 samples (**b**) ($n = 4$ paired patient samples). Log10-transformed FDR-adjusted $p$-values are on the y-axis and log2-transfomed fold change between time points is on the x-axis. Genes with the absolute log2-fold changes greater than 0.5 are shown in green, genes with a FDR-adjusted $p$-value were below 0.05 are shown in blue, and genes that meet both thresholds are in red. **c** Plot of significantly differentially expressed HALLMARK pathways (FDR-adjusted $p$-values < 0.05) for pair-wise comparisons between time points. Gene set statistics were run with fgsea using MSigDb[63] v7.4.1. Negative NES scores (blue) indicate pathways that are downregulated, while positive NES score (red) indicates pathway upregulation in C1D1 or C2D1. Source data are provided as a Source Data file. *NES, normalized enrichment score.*

immune responses remains relatively unclear and certainly deserves further attention. Recent seminal studies showed that intratumoral B-cells and associated tertiary lymphoid structures correlate with responses to checkpoint blockade immunotherapy and are prognostic for overall survival of cancer patients [47,48]. Other studies have investigated the correlation between circulating B cells and clinical response to anti-PD1 immunotherapy and showed a correlation between decreased B cells in the periphery and clinical response to immunotherapy[49,50]. These data support that B-cells represent a heterogenous population whose function spans antigen-presenting cells, regulatory cells, memory populations, and antibody-producing plasma cells[51,52]. Further characterization of B cell subpopulation and in-depth mechanistic effects will be crucial to understanding if and how entinostat impacts specific subsets of B cells and how this could impact the adaptive immune system response.

RNA-seq analysis demonstrated enhanced effects of combined entinostat plus nivolumab on a transcriptome-wide scale. Key molecular correlates include 1) upregulation of inflammasome-associated genes such as *IFN gamma*, and cytokines such as *CCXL9* and *CXCL10*; 2) modulation of checkpoint molecules; 3) upregulation of APC-associated MHCII antigen presentation genes and *CLEC9A* and *CLEC7A* 4) diminished tumor cell proliferation with associated downregulation of the c-MYC oncogene pathway. Collectively, these results reveal a myriad of transcriptional effects of the combinatorial therapy to stimulate immune signaling and to have potential direct effects on DC and APC function and indirect effects on T cells.

This study does have limitations. We noticed that many patients had progression of disease in the first month, when their drug exposure was predominantly entinostat. This is an important disease-based feature of PDA that needs to be kept in mind when lead-ins of drugs are

utilized that are not expected to have single agent efficacy. Accordingly, we noted an ORR of 11% vs the desired 15%, though our ORR was 17.6% in the 17 patients that at least reached cycle 2. One of our responders had a TMB of 13mut/Mb. As PDA with high TMB is rare, undertaking comparative effectiveness studies in this population is challenging. While a higher TMB shows promise as a predictive biomarker for patient selection in ICI treatment, multiple analyzes have not substantiated the hypothesis that a single TMB threshold can reliably identify patients across various cancer types who may benefit from ICI. Specifically, high TMB (defined as greater than 10 mut/mb) does not seem to correlate with an enhanced response rate or a response rate exceeding 20% in certain cancer types, such as PDA, which lack a clear association between neoantigen load and CD8 T-cell infiltration[53–56].

Second, the single-arm design of the current study limits the ability to definitively attribute clinical efficacy to the combination, but neither epigenetic agents nor anti-PD1 immunotherapy have previously demonstrated clinical responses in PDA[3]. Third, we did not observe significant changes in bulk RNA expression after the two weeks of entinostat single agent. We acknowledge that these agents may work better as sensitizers and that their dosage and biopsy timing must be wisely timed to capture their biological effects. Given the aggressiveness of the disease treated, we chose a short window for entinostat monotherapy lead-in in our trial to avoid delay of the combinatorial therapy. Fourth, the sample size of the correlative analyzes was limited by the 6-week biopsy being added at a time point during the latter 1/3 of the recruitment period. Finally, due to the limited sample size, we did not have enough homogeneity to evaluate treatment effects in responders vs. non-responders (and limited tissue in responders). Finally, one limitation of our analysis pertains to the absence of neighborhood-level examination. The scope of our study did not encompass an in-depth investigation into neighborhood dynamics, which could have provided valuable insights into contextual factors influencing our findings. The rapidly expanding knowledge and application of spatial single-cell tumor analysis will also help future analyzes of the T cell–DC-macrophages interaction, which needs to be harnessed for better invigoration of anti-tumor immune responses and possible therapeutic interventions.

We have ongoing studies to further investigate this time course in our preclinical models and to validate the effects of entinostat captured at additional time points using single-cell methods. The window of opportunity studies and studies in earlier stage disease could mitigate this problem. In addition, due to the nature of bulk-RNA sequencing and the variation in sample composition, we note the need for future work on the deconvolution of the gene expression profiles based on these identified cell proportions. Furthermore, patient-sourced, in vitro organoid system will be utilized and whole exome sequencing and bulk RNA sequencing are undergoing to investigate the hypothesis that HDAC inhibition can alter the neoantigen landscape, promoting the expression of cancer neoantigens and expanding neoantigen-specific responses by the host immune system.

In summary, this clinical trial tested the combination of HDAC and immune checkpoint inhibition in PDA patients. It provided initial evidence of the combination's safety, and while negative, it displayed a meaningful partial and durable objective response in a heavily pretreated subset of patients. These data are consistent with data reported by other groups that HDAC inhibitors have a role in combination with anti-PD1 therapy to increase benefit in both primary and secondary immune-resistant cancers. We validated previous preclinical data that HDAC inhibition favorably modulates the PDA TME, creating a less immunosuppressive milieu, including altering TAM populations, favoring Th1-like phenotypes, fomenting maturation of DCs, and increasing activation and proliferation of memory T cells in the periphery. These changes in the TME are concomitant with observations of myeloid activation and expansion systemically. This study creates a

roadmap for this strategy and future areas of exploration include additional mechanistic studies that will lead to combinatorial approaches, both preclinically and clinically, to expand the subset of patients that could benefit from an HDAC inhibitor-based immunomodulatory strategy.

## Methods

### Ethics and compliance
All procedures were conducted in accordance with the Declaration of Helsinki and the International Conference on Harmonization Guidelines for Good Clinical Practice. The protocol and all amendments were reviewed by the scientific review committee and institutional review board at the Johns Hopkins Sidney Kimmel Cancer Center. All patients provided written informed consent before enrollment.

### Trial registration
This trial is registered under the name 'A Clinical Trial of Entinostat in Combination with Nivolumab for Patients with Previously Treated Unresectable or Metastatic Cholangiocarcinoma and Pancreatic Adenocarcinoma' (registration no. NCT03250273r https://clinicaltrials.gov/study/NCT03250273?term=J1798&rank=1). Trial pre-registration: 8/14/2017. Study Start: 11/06/2017. No deviations since we were registered prior to initiation of the trial.

### Patient selection
Eligibility criteria included: patients at least 18 years old; histologically or cytologically confirmed pancreatic adenocarcinoma after progressing on one or two lines of therapy; ECOG PS ≤ 1; life expectancy greater than 12 weeks; adequate hematologic function (absolute neutrophil count ≥1500 cells per µl; white blood count ≥3000 cells per µl; platelets, ≥100,000 per µl; hemoglobin, ≥9 g l); adequate renal function (serum creatinine within the institutional upper limit of normal or creatinine clearance ≥60 ml min); adequate hepatic function (serum total bilirubin ≤1.5× the upper limit of normal, AST and ALT ≤ 3× the upper limit of normal or ≤5× the upper limit of normal in patients with liver metastases); participants were also required to have measurable disease and accessible non-bone tumor lesions for serial core biopsies

Exclusion criteria include participants who had received chemotherapy, radiotherapy, or surgery study 3 weeks of protocol treatment or (2 weeks for target therapy) and those who had not recovered (grade ≤1 or at baseline) from AEs due to agents administered more than 3 weeks earlier, except for alopecia. Participants who had previously received epigenetic therapy (i/e other HDAC inhibitors) or prior anti-PD-1, anti-PD-L1, anti-PD-L2, anti-CTLA4 antibodies or any other antibody or drug specifically targeting T cell costimulation or immune checkpoint pathways. Participants currently receiving any other investigational agents; those with known or suspected autoimmune diseases other than vitiligo, type 1 diabetes mellitus, residual hypothyroidism due to an autoimmune condition only requiring hormone replacement, psoriasis not requiring systemic treatment, or conditions not expected to recur in the absence of an external trigger condition. Participants requiring systemic treatment with either corticosteroid (the daily equivalent of >10 mg prednisone) or other immunosuppressive medications within 14 d of study drug administration).Participants with a known history of active TB, HBV, HCV, or HIV; uncontrolled intercurrent illness or psychiatric illness or social situations that would limit compliance with study requirements; pregnant or breastfeeding; known additional malignancy that is progressing or requires active treatment (except for non-melanotic skin cancer or carcinoma-in-situ of any type). Participants with a known history of or any evidence of symptomatic interstitial lung disease or any findings that may interfere with the detection or management of suspected drug-related pulmonary toxicity. Participants with an active infection requiring systemic therapy; received a live vaccine within 30 days of the planned start of study therapy; history of allergy to

study drug components; history of severe hypersensitivity reaction to any monoclonal antibody; uncontrolled brain metastases (patients treated with radiation ≥4 weeks prior with follow-up imaging showing control were eligible). Patients requiring concurrent administration of valproic acid; patients with evidence of clinical or radiographic ascites; participants who have had evidence of active or acute diverticulitis, intra-abdominal abscess, or GI obstruction and those with any contraindication to oral agents or significant nausea and vomiting, malabsorption, or significant small bowel resection that would preclude adequate absorption.

The trial protocol is provided in the Supplementary Note

## Study design and treatment

This was an open-label, single-arm, phase 2 clinical trial conducted at the Sidney Kimmel Comprehensive Cancer Center at Johns Hopkins, Baltimore, Maryland. Patients were enrolled between November 6, 2017 and November 5, 2020. Patients receive entinostat 5 mg orally once a week. After a fourteen-day lead-in with entinostat monotherapy, patients begin to concurrently receive entinostat 5 mg orally once a week plus nivolumab 240 mg every two weeks. Nivolumab was administered first as a 30-min i.v. infusion. After 4 months, therapy was continued with entinostat 5 mg weekly plus nivolumab at a dose of 480 mg, fixed dose every 4 weeks (maintenance phase) until disease progression (defined according to Response Evaluation Criteria in Solid Tumors (RECIST), version 1.1[57], unacceptable toxicity or withdrawal. Dose interruptions and management of immunologic toxicities were in accordance with the protocol. Safety was continuously monitored for unacceptable toxicities. unacceptable toxicity or withdrawal. Dose interruptions and management of immunologic toxicities were in accordance with the protocol. Safety was continuously monitored for unacceptable toxicities.

## Correlative science

All patients underwent fresh tumor biopsy (subsequent passes flash frozen, formalin-fixed paraffin-embedded, and preserved in RNAlater) and had research blood drawn at baseline, after two weeks of entinostat lead-in therapy (C1D1), and on week 6 following entinostat given with nivolumab combination therapy (C2D1). An optional biopsy was performed at the time of disease progression.

## Assessments

Participants were seen every 2 weeks for clinical assessments, including a physical examination for vital signs, performance status (PS), hematology, and biochemistry tests on or within 72 hours before day 1 and day 15 for the first 4 months. During the maintenance phase, patients were seen monthly.

Participants were evaluated for radiographic response every 8 weeks for the first 6 months and then every 12 weeks subsequently. Scans could also be obtained every 12 weeks at the clinician's request.

Patients were followed for survival until death, withdrawal of consent for follow-up, or up to 2 years. All AEs were monitored from registration until 30 days after treatment and were graded according to the National Cancer Institute Common Terminology Criteria for Adverse Events, version 4.03. SAEs that occurred within 100 days of the last treatment or before initiation of a new antineoplastic treatment were also followed and recorded.

Following disease progression, patient follow-up took place within 28 days from the last dose (±7 days), with subsequent follow-up (by phone or email) every three months (±2 weeks) for up to 2 years or study closure to monitor overall survival. Information on other cancer therapies after discontinuation from the study treatment was collected.

## Study endpoints

The primary trial endpoint was the objective response rate (ORR) assessed by RECIST 1.1. Secondary endpoints include safety, progression-free survival (PFS), and overall survival (OS). Exploratory objectives included biomarker and immunological analysis of serial tumor biopsies and peripheral blood samples.

Overall response rate and survival analyzes. A Simon's two-stage, minimax design was used to test the null hypothesis that the true ORR is 5% or less (not considered clinically compelling for this combination). In the first stage, 13 subjects accrued. If there were no responders among the first 13 subjects, the study was terminated for futility. Otherwise, 14 additional subjects were accrued to target 27 treated and response evaluable subjects. The null hypothesis was rejected if 4 or more responses were observed in 27 subjects. The probability of stopping the trial early for futility was 51% if the true ORR was 5% or less. This design yielded 80% power at a one-sided type I error rate of 5% when the true response rate was 20%[58]. An exact binomial test was used to evaluate the primary question of whether the response rate for combination therapy exceeds the historical rate (5%) for the single agent. Response rates were reported with exact confidence intervals. The median time to event was calculated and reported with confidence intervals. PFS was defined as the time until the earlier date of either PD or death or was otherwise censored at the date of the last follow-up. OS was defined as time to death from any cause or otherwise censored at the date of the last follow-up for patients still alive at the time of analysis. OS curves, OS medians with 95% CIs, and OS rates at 6, 12, 24, and 36 months with 95% CIs were estimated using the Kaplan–Meier methodology. PFS curves, PFS medians with 95% CIs, and PFS rates at 6, 12, and 24 months with 95% CIs were estimated using Kaplan–Meier methodology. Duration of response (DOR) was calculated for subjects who achieved the best overall response of CR or PR with entinostat in combination with nivolumab. For such subjects, the duration of response was defined as the number of weeks from the start date of PR or CR (whichever response was recorded first) and subsequently confirmed to the first date that recurrent or progressive disease or death was documented. DOR was summarized descriptively.

The primary analysis was based on the intention to treat (ITT) population, while the per-protocol analysis was defined in patients who received at least one restaging scan

## CyTOF data acquisition and analysis

All staining procedures were carried out as previously described[59]. Commercially available and custom-conjugated antibodies were cocktailed into panels as detailed in Supplementary Data 22–24. Samples were first live-dead stained using palladium (Sigma) for 5 minutes at RT and immediately quenched with media. For multiplexing, 10 samples were barcoded using CD45 antibodies tagged with 5 different isotopically enriched metals based on a 5-choose-3 scheme, 10 min at RT. Upon batching, Fc receptors were blocked (Invitrogen). Samples were then stained with antibody cocktails at their indicated dilutions using two panels of markers: one is myeloid cell oriented and the other is lymphoid cell oriented (Supplementary Data 22–24). Chemokine receptors were first stained at 37 °C for 10 min, then all other surface markers at RT for an additional 20 min, and finally followed by intracellular markers after a Cytofix/Cytoperm Kit (BD Biosciences) step as per the manufacturer's protocol. Completely stained cells were fixed with 1.6% paraformaldehyde (ThermoFisher) in PBS and stored for up to 1 week at 4 °C. All cells were then stained with rhodium Cell-ID (Fluidigm) prior to acquisition. All data were acquired using Helios™ at the Sidney Kimmel Comprehensive Cancer Center in Baltimore, Maryland. For preprocessing of CyTOF data, randomization, bead-based normalization, and bead removal of data were performed in CyTOF software (Fluidigm®) v6.7 followed by gating of cell events (rhodium vs. cell length signal) that are viable (106 Pd vs. 108 Pd) in FlowJo (BD Biosciences) v10.5. Individual samples were debarcoded by hierarchal gating (three positive and two negative CD45 axes) and exported as separate FCS files for analysis. For all CyTOF analyzes, a

computational pipeline based on diffcyt2 was employed using R. For unsupervised clustering, the FlowSOM3 algorithm was used to identify meta-clusters that were then annotated and merged into final cell subtypes based on published literature. To compare the cell proportions and protein expression distributions in different clusters between time points, we used pairwise two-sided paired Wilcoxon test.

## mIHC

Multiplex IHC (mIHC) is a validated sequential staining technique coupled with a robust computational workflow culminating with image cytometry focused on auditing immune contexture in FFPE tissue specimens[60](Supplementary Fig. 2a). This approach is an efficient way to analyze small biopsy specimens. The antibody panel employed for this study has been previously published and includes 23 antibodies in addition to hematoxylin nuclear staining for single-cell segmentation. Hierarchical image cytometry gating allows for the logical combination of markers for the identification of 10 cell lineages including neoplastic epithelium, CD4+ and CD8 + T cells, CD20 + B cells, macrophages, monocytes, DCs, and granulocytes. In addition, 7 markers are included for the identification of cellular states, for example, Ki67+ (proliferative), Granzyme B+ (cytotoxic), and immune regulatory (PD-1, PD-L1) (Supplementary Fig. 2b).

## RNA-seq

Patients underwent biopsy at baseline ($n = 25$), at C1D1 (after two weeks of entinostat lead-in therapy, $n = 21$), and at C2D1 (at week 6 following entinostat given with nivolumab, $n = 4$). A total of 5 samples were removed from analysis due to insufficient amount of the available RNA ( < 50 ng total) required for RNA-seq (2 baseline and 3 C1D1 samples) (Supplementary Data 25). Each cryovial containing a single core biopsy sample in RNAlaterTM was placed in the refrigerator (4 °C) for ≥16–24 h after the biopsy procedure. After ≥16–24 h, the RNAlaterTM vials were stored at −70 °C or below for storage. For RNA sequencing experiments, RNA libraries were generated using the TruSeq Stranded Total RNA Library kit according to manufacturer instructions (Illumina; San Diego, CA). Quality and quantity of the resulting cDNA were monitored using the Bioanalyzer High Sensitivity kit (Agilent). mRNA libraries were sequenced on an Illumina Novaseq 6000 instrument using 150 bp paired-end dual indexed reads and 1% of PhiX control on an S4 flowcell. Depth of coverage was targeted to a total of 50 million reads per library. Illumina's CASAVA (v1.8.4) with default parameters was used to generate FASTQ files with reads. Then the reads were trimmed with trimgalore (v0.6.3) with default parameters and aligned to human genome (hg38) and quantified with the rsem algorithm (v1.3.0)[61]. The RSEM expected counts were used for gene level expression. We evaluated sequencing quality from the distribution of expected counts as visualized in a boxplot of log counts. We observed no samples with zero median expression, reflective of a low read count, so all samples had good quality. We used principal component analysis (PCA) of the variance stabilization transform (vst) RNA-seq data to evaluate sample clustering. For differential expression analysis, we completed a paired analysis between time points using the DESeq2 package (v1.32.0)[62], including the patient as a covariate in our model, along with our comparison of interest. Estimated fold changes are shrunk with ashr using lfcShrink[63] to account for the variation in the samples in this dataset. Genes were statistically significant if the absolute log2-fold changes after shrinkage were greater than 0.5 and the FDR-adjusted $p$-value was below 0.05 (Supplementary Data 16–18). Gene set statistics were run with fgsea using MSigDb[63] v7.4.1 pathways annotated in the HALLMARK databases. Gene sets were considered significantly enriched for FDR-adjusted $p$-values below 0.05 (Supplementary Data 19–21), and the results were visualized with ggplot2[64].

## Cytokines analysis

The concentrations of cytokines and chemokines were assessed by the Sidney Kimmel Comprehensive Cancer Center (SKCCC) Immune Monitoring Core using Luminex bead-based immunoassays. The Bioplex 200 platform (Biorad, Hercules CA) was used to determine the concentration of multiple target proteins in serum/plasma specimens following Immune Monitoring Core SOPs, and concentrations were determined using 5 parameter log curve fits (using Bioplex Manager 6.0) with vendor-provided standards and quality controls. The HCYTA-60K-PX48 panel (Millipore) was used to detect (sCD40L, EGF, Eotaxin, FGF2, FLT-3L, Fractalkine, GCSF, GMCSF, GROα, IFNα2, IFNγ, IL-1α, IL-1β, IL-1RA, IL-2, IL-3, IL-4, IL-5, IL-6, IL-7, IL-8, IL-9, IL-10, IL-12(p40), IL-12(p70), IL-13, IL-15, IL-17A, IL-17E/IL-25, IL-17F, IL-18, IL-22, IL-27, IP-10, MCP-1, MCP-3, MCSF, MDC, MIG/CXCL9, MIP-1α, MIP-1β, PDGF-AA, PDGF-AB/BB, RANTES, TGFα, TNFα, TNFβ, VEGF-A) in the serum/plasma specimens (Supplementary Data 11).

## Statistical considerations

Multiple test adjustment is not considered due to small sample size and the exploratory nature of the analyzes. Unadjusted $p$-values are reported to all analyzes except for RNA-seq. Statistical tests with $p$-value less than 0.05 are considered significant. All statistical analyzes were performed in R version v4.0.2

## Reporting summary

Further information on research design is available in the Nature Portfolio Reporting Summary linked to this article.

## Data availability

The raw RNA sequencing data are available under restricted access in dbGaP repository under accession code phs003615.v1.p1 [https://www.ncbi.nlm.nih.gov/projects/gap/cgi-bin/study.cgi?study_id=phs003615.v1.p1]. The RSEM expected gene counts are publicly available in GEO database under accession code GSE248014. The CyTOF data are publicly available in Zenodo under [https://doi.org/10.5281/zenodo.12802613]. The study protocol is available in the Supplementary Information file. The authors declare that the minimal data set and source data for clinical data for this study cannot be shared publicly due to ethical and legal restrictions on sharing de-identified data that aligns with the consent of research participants. Current JHU compliance policies require data with no direct consent for public open access sharing be under restricted access. We provide access through dbGAP, an established repository for clinical data that provides open access without a fee restricted to approved researchers under a Data Use Agreement. JHU compliance policy for dbGAP requires additional anonymization of certain demographics, including the use of age ranges and limiters to outlier values for weight, height, and certain rare diseases, while retaining sufficient value for reference and validation of results. Researchers can request more detailed data from the corresponding author shared through an approved collaboration arrangement. The remaining data are available within the Article, Supplementary Information or Source Data files. Source data are provided with this paper.

## Code availability

The code associated with this manuscript is available on Zenodo: https://zenodo.org/doi/10.5281/zenodo.12795850. The mIHC analysis protocol and code are available on protocols.io: https://doi.org/10.17504/protocols.io.n92ldmmznl5b/v2.

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

## Acknowledgements

We thank the Experimental and Computational Genomics Core members at SKCCC including Dixie Hoyle, Jennifer Meyers, Kornel Schuebel, and Hai Xu for generating RNA-seq data. This study was supported by the Lustgarten Foundation's Research Investigator's Award Program, NIH P01-CA247886-01A1, 2018 Conquer Cancer Young Investigator Award, MD Anderson Cancer Center SPORE in Gastrointestinal Cancer – The Career Enhancement Program, and NCI R50CA243627 grant (LD). The Cancer Center Grant - P30CA006973. Daria A. Gaykalova was supported by a Research Scholarship Grant, RSG-21-020-01-MPC from the American Cancer Society, and R01DE027809 from the National Institute of Health. This work was made possible in part through the support of the Maryland Cancer Moonshot Research Grant to the Johns Hopkins Medical Institutions (FY24). We thank BMS and Syndax for supplying study medications. The sponsor did not have a direct role in study design, data collection, and analysis, or manuscript writing.

## Author contributions

M.B., N.S.A and E.J. contributed to the conception, design, and planning of the study. M.B., L.D., J.N.D., L.C., D.N.S., J.A.T., S.C., N.G., A.H., W.J.H., C.T., R.W., J.L., S.M., B.C., N.S.A. acquired the data. M.B., L.D., J.N.D., C.B.B., L.C., D.N.S., J.A.T., S.C., N.G., A.H., W.J.H., C.T., R.W., J.L., S.M., B.C., A.S., D.A.G,, S.Y., E.J.F., L.M.C., M.Y., E.J. and N.S.A. contributed to the interpretation of the results. M.B., M.Y. and N.S.A. drafted the manuscript. M.B., L.D., J.N.D collaborated on the manuscript tables and figures. All authors contributed to the drafting or critical review of the manuscript and all authors had access to the data, which was verified by M.B., N.S.A., L.D., J.N.D., L.M.C.

## Competing interests

M.B. served on advisory boards for AstraZeneca, Incyte. C.B.B is an employee of, and holds equity in, Akoya Biosciences, Inc. W.J.H. reports patent royalties from Rodeo/Amgen; grants from Sanofi, NeoTX, and Ribosience; speaking/travel honoraria from Exelixis and Standard Bio-Tools. MY receives grant/research support (to Johns Hopkins) from Bristol-Myers Squibb, Exelixis, Incyte, and Genentech; receives honoraria from Genentech, Exelixis, AstraZeneca, Replimune, Hepion, Lantheus; MY is the co-inventor of patents pertaining to cancer vaccines and is a cofounder with equity of Adventris Pharmaceuticals, all outside of the submitted work. E.J.F is on the scientific advisory board for Resistance Bio, has consulted for Merck and Mestag Therapeutics, and receives research grants from Roche/Genetech and Abbvie Inc. L.M.C. has received reagent support from Cell Signaling Technologies, Syndax Pharmaceuticals, Inc., ZielBio, Inc., and Hibercell, Inc.; holds sponsored research agreements with Syndax Pharmaceuticals, Hibercell, Inc., Prospect Creek Foundation, Lustgarten Foundation for Pancreatic Cancer Research, Susan G. Komen Foundation, and the National Foundation for Cancer Research; is on the Advisory Board for Dispatch Biotherapeutics, Carisma Therapeutics, Inc., CytomX Therapeutics, Inc., Shasqi, Kineta, Inc., Hibercell, Inc., Cell Signaling Technologies, Inc., Alkermes, Inc., Raska Pharma, Inc., NextCure, Guardian Bio, AstraZeneca Partner of Choice Network (OHSU Site Leader), Genenta Sciences, Pio Therapeutics Pty Ltd., and Lustgarten Foundation for Pancreatic Cancer Research Therapeutics Working Group, Inc. E.J. reports other support from Abmeta, other support from Adventris, personal fees from Achilles, personal fees from DragonFly, non-financial support from Parker

Institute, personal fees from Surge, grants from Lustgarten, grants from Genentech, personal fees from Mestag, personal fees from Medical Home Group, non-financial support from BMS, grants from Break Through Cancer, personal fees from CPRIT, personal fees from Neuvogen, non-financial support from HDT Bio, and personal fees from NeoTx outside the submitted work. N.S.A. receives grant/research support (to Johns Hopkins) from Bristol-Myers Squibb. L.D., J.N.D., L.C., D.N.S., J.A.T., S.C., N.G., A.H., C.T., R.W., J.L., S.M., B.C., A.S., D.A.G., S.Y. declare no competing interest related to this study.

## Additional information

**Marina Baretti**[1], **Ludmila Danilova** [1], **Jennifer N. Durham**[1], **Courtney B. Betts**[2,3], **Leslie Cope**[1], **Dimitrios N. Sidiropoulos** [1,4,5], **Joseph A. Tandurella** [1], **Soren Charmsaz** [1], **Nicole Gross** [1], **Alexei Hernandez**[1], **Won Jin Ho** [1], **Chris Thoburn** [1], **Rosalind Walker**[1], **James Leatherman**[1], **Sarah Mitchell**[1], **Brian Christmas**[1], **Ali Saeed**[1], **Daria A. Gaykalova**[1,6,7], **Srinivasan Yegnasubramanian** [1,4,5,8], **Elana J. Fertig** [1,4,5,9,10], **Lisa M. Coussens** [2], **Mark Yarchoan** [1,4,5], **Elizabeth Jaffee** [1,4,5] & **Nilofer S. Azad**[1,4,5] ✉

[1]Department of Oncology, Sidney Kimmel Comprehensive Cancer Center, Johns Hopkins University, Baltimore, USA. [2]Department of Cell, Developmental & Cancer Biology and Knight Cancer Institute, Oregon Health & Science University, Portland, USA. [3]Akoya Biosciences, Marlborough, USA. [4]The Convergence Institute, Johns Hopkins University, Baltimore, USA. [5]Bloomberg-Kimmel Institute at Johns Hopkins, Baltimore, USA. [6]Department of Otorhinolaryngology-Head and Neck Surgery, Marlene & Stewart Greenebaum Comprehensive Cancer Center, University of Maryland Medical Center, Baltimore, USA. [7]Institute for Genome Sciences, University of Maryland School of Medicine, Baltimore, USA. [8]Johns Hopkins in Health Precision Medicine, Johns Hopkins Medicine, Baltimore, USA. [9]Department of Applied Mathematics and Statistics, Johns Hopkins University, Baltimore, USA. [10]Department of Biomedical Engineering, Johns Hopkins University, Baltimore, USA. ✉e-mail: nazad2@jhu.edu

