## [Peer Review File · Nature Communications]

Entinostat in combination with nivolumab in metastatic pancreatic ductal adenocarcinoma: a phase 2 clinical trialEditorial note: This manuscript has been previously reviewed at another journal that is not operating a transparent peer review scheme. This document only contains reviewer comments and rebuttal letters for versions considered at Nature Communications.

REVIEWERS' COMMENTS

Reviewer #1 (Remarks to the Author):

The authors have addressed my comments and suggestions sufficiently. I highly recommend publication of this study, which reports an impressive response in 10% of the heavily pretreated and metastatic PDAC patients of this study.

Reviewer #5 (Remarks to the Author):

This is a single institution phase 2 trial evaluating combinatorial HDAC inhibitor (entinostat) with anti PD1 antibody in patients with locally advanced or metastatic, treatment refractory pancreas cancer. The investigation of this combinatorial regimen in PDAC is supported by the authors' preclinical work suggesting that HDAC inhibitor shifted the MDSC population to a less immunosuppressive subtype in the tumor microenvironment and combinational entinostat with PD-1 antibody improved murine survival. compared to monotherapy. The manuscript is well written. Although this combination has been evaluated in other solid tumors (ie breast cancer), this is the first study to evaluate this regimen in a pancreas cancer population where novel therapies are sorely needed.

The study failed to meet its primary endpoint but did demonstrate an ORR in 3 patients who have been refractory to multiples line of therapy. Immunologic assessments to better understand mechanisms of response and resistance to therapy are limited and unclear given the small number of responses and limited tissue samples particularly in the responders. The original conclusions appeared overstated however the authors have addressed this in the revised version and adequately addressed reviewer comments. The methodology is sound and methods are appropriate and thorough. Ultimately, the next steps for this combination remain unclear and the authors conclude that this study portends further

mechanistic preclinical and clinic studies to understand who will benefit from this combinatorial regimen.

Reviewer #6 (Remarks to the Author):

(Statistical review)

1- Estimates and confidence intervals on date collected with Simon's two-stage design would need a correction, was it applied?

2- Line 258, there is a typo: It should refer to Supplementary Fig 2b.

3- Supplementary Fig 3 and Figures 7 a and b: the analysis in Supplementary Fig 3 is based on 16 paired samples, while in Figures 7 a and b only on a (subset?) of 4 paired samples. It is likely that these 4 remaining patients were systematically different from those who did not survive. I think that the differences in C1D1 vs Baseline and C2D1 vs Baseline could be more robustly interpreted if (additionally to the current analysis) the same 4 samples were used for both comparisons.

“Entinostat in combination with nivolumab in metastatic pancreatic ductal adenocarcinoma: a phase 2 clinical trial”

We have modified our manuscript to optimize our report based on the review. Below, we address the points raised and other relevant sub-comments.

Reviewer #1 (Remarks to the Author):

The authors have addressed my comments and suggestions sufficiently. I highly recommend publication of this study, which reports an impressive response in 10% of the heavily pretreated and metastatic PDAC patients of this study.

Reviewer #5 (Remarks to the Author):

This is a single institution phase 2 trial evaluating combinatorial HDAC inhibitor (entinostat) with anti PD1 antibody in patients with locally advanced or metastatic, treatment refractory pancreas cancer. The investigation of this combinatorial regimen in PDAC is supported by the authors' preclinical work suggesting that HDAC inhibitor shifted the MDSC population to a less immunosuppressive subtype in the tumor microenvironment and combinational entinostat with PD-1 antibody improved murine survival. compared to monotherapy. The manuscript is well written. Although this combination has been evaluated in other solid tumors (ie breast cancer), this is the first study to evaluate this regimen in a pancreas cancer population where novel therapies are sorely needed.

The study failed to meet its primary endpoint but did demonstrate an ORR in 3 patients who have been refractory to multiples line of therapy. Immunologic assessments to better understand mechanisms of response and resistance to therapy are limited and unclear given the small number of responses and limited tissue samples particularly in the responders. The original conclusions appeared overstated however the authors have addressed this in the revised version and adequately addressed reviewer comments. The methodology is sound and methods are appropriate and thorough. Ultimately, the next steps for this combination remain unclear and the authors conclude that this study portends further mechanistic preclinical and clinic studies to understand who will benefit from this combinatorial regimen.

Reviewer #6 (Remarks to the Author):

(Statistical review)

1- Estimates and confidence intervals on data collected with Simon's two-stage design would need a correction; was it applied?

Response: Reported response rates and confidence intervals were not adjusted for Simon's two-stage design. In order to keep our results comparable with the reports of other studies that used Simon's two-stage design but reported the naïve estimate, we would prefer to retain the original results, but will follow journal policy and the editor's preference.

2- Line 258, there is a typo: It should refer to Supplementary Fig 2b.

Response: Corrected.

3- Supplementary Fig 3 and Figures 7 a and b: the analysis in Supplementary Fig 3 is based on 16 paired

samples, while in Figures 7 a and b only on a (subset?) of 4 paired samples. It is likely that these 4 remaining patients were systematically different from those who did not survive. I think that the differences in C1D1 vs Baseline and C2D1 vs Baseline could be more robustly interpreted if (additionally to the current analysis) the same 4 samples were used for both comparisons.

Response: Thank you for your insightful comments regarding the analysis presented in Supplementary Figure 3 and Figures 7 a and b. We appreciate your attention to the details of our methodology and the implications of our sample selection. We would like to clarify that a third timepoint for biopsy (C2D1) was added as an amendment to our study protocol after more than half of the initial patient cohort was already enrolled. Consequently, only a minority of patients consented in the first place to undergo biopsies at this additional time point. On note, this amendment was implemented after the enrollment of the patients who demonstrated a response to the therapy. As a result, the four patients who had biopsies at the additional timepoint do not necessarily represent a biologically different population; instead, they reflect the change in protocol rather than an inherent difference in patient characteristics or response to treatment. Given this context, the small sample size, and the lack of observed response in these four patients, we determined that performing a post hoc analysis would introduce bias and potentially lead to misleading conclusions.